# Spatio–Temporal Pattern of the Urban System Network in the Huaihe River Basin Based on Entropy Theory

**DOI:** 10.3390/e21010020

**Published:** 2018-12-27

**Authors:** Yong Fan, Renzhong Guo, Zongyi He, Minmin Li, Biao He, Hao Yang, Nu Wen

**Affiliations:** 1Research Institute for Smart Cities & Shenzhen Key Laboratory of Spatial Information Smart Sensing and Services, School of Architecture and Urban Planning, Shenzhen University, Shenzhen 518061, China; 2College of Geographic Sciences, Xinyang Normal University, Xinyang 464000, China; 3School of Resource and Environmental Sciences, Wuhan University, Wuhan 430079, China

**Keywords:** urban system, gravity model, spatial interaction, information entropy, Huaihe River Basin

## Abstract

As complex systems, the spatial structure of urban systems can be analyzed by entropy theory. This paper first calculates the interaction force between cities based on the gravity model, the spatial relationship matrix between cities is constructed using the method of network modeling, and the spatial network modeling of urban system can be calculated. Secondly, the Efficiency Entropy (EE), Quality Entropy (QE), and System Entropy (SE) of urban system network are calculated and analyzed by information entropy. Finally, taking the Huaihe River Basin (HRB) as a case study, model verification and empirical analysis are performed. It is found that the spatio–temporal pattern of the urban system network structure in the basin is uneven: in space, the urban system network in the HRB presents a layer-by-layer spatial distribution centered on the core city of Xuzhou; meanwhile, the overall urban system network in the basin presents an orderly development trend. This study has certain theoretical and practical value for the planning of urban and urban systems and the coordinated development of regions.

## 1. Introduction

Urban systems are the network systems of urban areas and its complex land [1]. Urban systems are carriers of human activities, and the changes of spatio–temporal pattern of urban systems contain the interactive process between humans and nature. Studying the spatio–temporal pattern of regional urban systems is conducive to the scientific understanding of the structure and evolution of urban systems.

In the course of a city’s development, the exchange of material, energy, and information will inevitably take place with its surrounding environment, which forms the interaction among cities [2] and promotes the formation of regional urban system network. Urban system networks are complex networks affected by economic, political, cultural, and other factors. The model and spatial analysis have always been a hot issue in urban geography. The Theory of Central Place [3], Point-Axis Theory [4], Dual-Nuclei Structural Model [5], City Network [6], Network City [7] and City tree [8,9,10], showed a networked trend of the research on the spatial complexity of urban systems. Analysis based on different spatial scales using network theory showed that with the increase of the complexity of urban system structure, its structure becomes more effective and stable, and more conducive to the sustainable development of the urban system [11]. Gravity Model [12,13], Social Network [14,15], Resistance Model [9], self-organizing structure model [16] and systematic dynamics [17] have been used to describe spatial interaction of urban system.

The physical concept of entropy can be used to measure system complexity and balance [16]. As a dissipative network structure, the spatial structure of an urban system can be analyzed by entropy theory. Many different entropic measures have been developed in the context of urban study, for example, the “entropy law” of urban land use [18], the information entropy of urban system [19], maximum information entropy assessment of urbanization quality [20,21], and built-up area [22]. Additionally, multiscale entropy [23] and the entropy value method [24] are widely applied to the study of the spatial structure of urban systems. The use of entropy in urban and regional modeling has introduced a new framework for constructing spatial interaction and associated location models [25]. However, there has been limited research on the entropy of spatio–temporal pattern of urban systems. Rarely, studies have adopted a network entropies measurement [26] to quantify the robustness of the evolving urban system, and proposed a statistical visualization model to quantify the spatio–temporal pattern of regional urban system at different scales. The efficiency and quality of the spatial structure of urban system are needed to quantify the evolution of regional system.

In this paper, the spatial network of an urban system is constructed using the gravity model. Based on the information entropy theory, the Efficiency Entropy (EE), Quality Entropy (QE), and System Entropy (SE) of the spatial network of urban system are calculated and analyzed. We try to use entropy theory to analyze the spatio–temporal pattern of urban network evolution in efficiency and quality. Finally, as a complete natural geographical unit, the Huaihe River basin (HRB) is taken as a case study area to complete the experimental verification and result analysis. The entropy of urban system network is a comprehensive reflection of dynamic changes and degrees of transformation of varied urban system of the study area in a certain period, and, to some degree, can guide the optimization and adjustment of the spatial structure of regional urban system.

## 2. Methods and Data

### 2.1. Gravity Model

Newton’s Law explains the relationships between a force and the mass of objects within a given system, in which the power of interaction force of the objects can be measured [11] as shown in Equation (1):(1)$$F=G\frac{m_{1}m_{2}}{r^{2}}$$
where *F* is the interaction force between two objects *m*_1_ and *m*_1_, *G* is the result of a given gravitational constant, and *r* is the distance between the two objects.

Similar to the interaction between objects, the interaction between cities is also subject to a law of attraction. The “gravity model” was first introduced into the analysis of spatial interaction of urban system by Zipf in 1946. Since then, the gravity model has been widely used to study the distance attenuation effect and space interaction [15,17,27,28].The basic equation of the gravity model is:(2)$$F_{ij}=K\frac{M_{i}M_{j}}{D_{ij}{}^{\beta 1}}$$
where *F_ij_* is the interaction force between city *i* and city *j*, *M* represents the quality of the city, *D_ij_* indicates the euclidean distance between city *i* and city *j*, *K* is a constant, and *β*^1^ is the distance attenuation coefficient.

This study is to explore the spatial interaction between cities, which is mainly driven by economic development. Therefore, the characterization of urban “quality” should focus on measuring urban economic development level, so, GDP is the primary indicator of urban quality. Considering that people are the subject of urban activities and the executors of urban spatial interaction, population size should also be one of the important indicators to measure the “quality” of cities. Empirical studies show that the strength of inter-city links is inversely proportional to the square of the distance between two cities, and the attenuation coefficient of distance is equal to 2 [17]. Considering the causality between the potential of economic links and the actual economic links, there is no equivalence between the two cities. Therefore, the empirical constant *K* is modified by the ratio of a city’s Gross Domestic Product (GDP) to the combined GDP of the two cities [17]. Thus, the gravity model equation is obtained as Equation (3):(3)$$F_{\mathrm{ij}}=K\times \frac{\sqrt{P_{i}G_{i}}\times \sqrt{P_{j}G_{j}}}{d^{2}},\;K=\frac{G_{i}}{G_{i}+G_{j}}$$
where *G_i_* and *G_j_* are the GDP of city *i* and city *j*, respectively, and *P_i_* and *P_j_* are the household registration populations who have registered permanent residence in the public security household registration management office of residence in accordance with the Regulations of the People’s Republic of China on Registration of Household Registration [29] of city *i* and city *j*, respectively.

### 2.2. Efficiency Entropy (EE)

The efficiency of a network structure refers to the speed of interaction between nodes, and the EE refers to the uncertainty in the process of interaction between nodes. The larger the EE value of the network structure, the more effective the spatial interaction between nodes in the network structure. The EE of the spatial network of an urban system indicates the validity of the interaction of matter, energy, and information between cities.

In a network structure, if two nodes can communicate with each other, they are called a connection, and the shortest path between the two cities is recorded as the length of the connection (*Lij*). The micro-state is the possible microscopic state of the network when analyze the network from different perspectives. *A*_1_ is the total number of information microstates of the network structure, $A_{1}=\sum_{i=1}^{N}\sum_{j=1}^{N}L_{ij}$. For any node, its probability of microcosmic realization is $P_{1}(ij)=L_{ij}/A_{1}$. The EE of any two nodes in the network structure is $H_{1}(ij)=-P_{1}(ij)\log_{2}P_{1}(ij)$. The total EE of a network structure is $H_{1}=\sum_{i=1}^{N}\sum_{j=1}^{N}H_{1}(ij)$, and its largest EE is $H_{1m}=\log_{2}A_{1}$. The order of network structure aging is $R_{1}=1-H_{1}/H_{1m}$ [30].

### 2.3. Quality Entropy (QE)

The quality of a network structure is a measurement of the accuracy of network node interaction. QE describes the size of node interaction uncertainty; the larger the QE value of a network structure, the better the spatial interaction between nodes in the network structure. The QE of the spatial network of an urban system is another measurement of the effectiveness of the interaction of matter, energy, and information between cities.

In a network structure, the number of nodes directly associated with a node is recorded as the connection span *K_i_* of the modified node. *K_i_* is the number of nodes in the network structure that are directly related to this node. It reflects the connection strength between network nodes. The larger *Ki* is the more complex the corresponding network structure is. The total number of microcosmic *A*_2_ of a network structure is the sum of all connection spans.

For any node, its probability of microcosmic realization is $P_{2}(i)=k_{i}/A_{2}$. The QE of any node in the network structure is $H_{2}(i)=-P_{2}(i)\log P_{2}(i)$, and the total QE of the network structure is $H_{2}=\sum_{i=1}^{N}H_{2}(i)$. The largest QE of the network structure is $H_{2m}=\log_{2}A_{2}$, its order of quality is $R_{2}=1-H_{2}/H_{2m}$ [30].

### 2.4. System Entropy (SE)

The SE of a network structure is derived from the EE and QE of the network structure, and the expression is $H=\alpha H_{1}+\beta H_{2}$, in which *α* and *β* represent the occupation ratio for EE and QE, respectively. In this study, we set *α* and *β* as 1. Thus, SE is $H=H_{1}+H_{2}$, and the order degree $R=R_{1}+R_{2}$ [30].

### 2.5. Data and Processing

#### 2.5.1. Study Area

The HRB is a transitional region from south to north for the Chinese urban system, and is located in mid-eastern China between the Yangtze River and the Yellow River basin (Figure 1). The HRB covers one provincial city and 27 prefecture-level cities. The total population of the HRB is 165 million, and the mean population density is 611 people per km^2^, which is 4.8 times the national average population density in China (122 people per km^2^), and the highest of the basin regions of all large rivers in China.

#### 2.5.2. Data Sources

Administrative division data: The administrative division data at a scale of 1:4,000,000 [31].Statistical data: The statistical data (population and GDP data) are taken from the Chinese Urban Statistical Yearbook from 2006, 2010, and 2014 [32,33,34].

#### 2.5.3. Data Processing and Analysis

The flow of data processing and analysis is shown in Figure 2. Firstly, in order to characterize urban quality, data were collected on the registered population and GDP of each city in 2006, 2010, and 2014, and the parameters of euclidian distance between cities were calculated using ArcGIS (Esri, Redlands, CA, USA). Secondly, based on the gravity model from Section 2.1, the interaction force (*F_ij_*) between cities was calculated. According to the interaction force of each city, the UCINET was applied to the calculation of the connection matrix between cities and the visual expression. According the connection matrix between cities and the visual expression, the appropritate threshold value of urban system network was determined. The spatial network structure of the urban system in the HRB was completed using the ArcGIS Platform based on the threshold value. Finally, the EE, QE, and SE from Section 2.2, Section 2.3 and Section 2.4 were used to analyze the spatial network structure of the urban system in the HRB.

## 3. Results

### 3.1. Urban System Network in the HRB

In order to facilitate the visual expression and analysis of the urban system network, the threshold value is 4—that is, the connection (the interaction force) value should be higher than 4, as shown in Figure 3. Connection values between cities of less than 4 are temporarily ignored.

From Figure 3, it can be seen that Xuzhou is the center of the whole urban system, but that the overall connection of the network is relatively loose. The urban system network of the HRB developed greatly from 2006 to 2014. In 2006, due to the low level of urban development and the low interaction among cities, the urban system network scale was small and concentrated in the eastern part of the HRB. However, with the acceleration of urban development and the more frequent interaction between cities, the urban system network gradually expanded. The gradual development shows that the internal network became more complex, and the external network expanded gradually. The urban system network had basically covered the entire basin by 2014.

### 3.2. Entropy of Urban System Network

#### 3.2.1. Efficiency Entropy (EE) of Urban System Network

Generally speaking, a high EE value indicates that information transmission efficiency maintains a higher level. This phenomenon is related to its higher level of economic development. With the continuous development and expansion of the urban system network, the information exchange between these original cities is gradually strengthened.

Efficiency Entropy (EE) of the urban system network of HRB from 2006 to 2014 is shown in Figure 4. The EE values of Zhengzhou, Kaifeng, and Heze were higher in 2010, and were still at a higher level in 2014, indicating that although these cities had developed, the efficiency of information exchange with other HRB cities was still low, which may be related to their geographical location. The EE values of Bozhou, Huainan, and Shangqiu were also low in 2014, but since 2014 the value has been at a relatively high level, indicating that although these points enter the network, the spatial interaction between these points and the cities in the whole river basin is not close enough.

In order to facilitate the macroscopic summary of the urban system spatial network in the HRB, the EE of the HRB network was summarized, as shown in Table 1. It can be seen from Table 1 that with the continuous development and expansion of the urban system network of the HRB, the total number of connections and the total micro-state were constantly increasing, and the EE value of the urban system network also experienced significant growth. The order degree of the urban system network was in a relatively stable state (0.17). This also shows that the efficiency of information transmission and communication in the HRB urban system network was in a stable state, and there was no information flow because of the expansion the of urban system network. With the development of the urban system network, the information exchange rate between the new nodes and the existing nodes was at a good level, and there was no communication barrier between the new nodes and the existing nodes.

#### 3.2.2. Quality Entropy (QE) of Urban System Network

Figure 5 shows the QE of each city in 2006, 2010, and 2014. The QE value of Xuzhou was the highest from 2006 to 2014 which show that Xuzhou is core for the urban system network in HRB. Zhengzhou, Yancheng, Suqian, Shangqiu, Rizhao, Pingdingshan, Luohe, Lianyungang Kaifeng, Jining, Huainan, Heze, Bozhou and Bengbuwere not present in the urban system network in 2006 and Bozhou was still not present in 2010 indicate that Bozhou was on the edge of the network. However, the entropy values of some cities were at a lower level in 2014 than 2006 and 2010, indicating that the spatial interaction between cities in the network became lower.

Table 2 shows the overall network QE situation, from the overall perspective of the network. The size of the city network in 2006 was small, with a large number of nodes joining the city network in 2010, the size of the city network tends to stabilize. The total micro-state increased in three periods, especially from 2006 to 2010. Correspondingly, the QE also increased to a certain extent. The fluctuation range of order degree is not obvious, which indicates the accuracy of network information transmission. Compared with 2006, the degree of order in 2010 decreased slightly, indicating that the overall structure of the urban system network was still affected by the large number of new nodes entering the urban system network. By 2014, with the development and improvement of the network, the degree of order had increased to a certain extent, indicating that the new nodes in 2010 were better.

#### 3.2.3. System Entropy (SE) of Urban System Network

Table 3 shows the SE of the urban system network. It was continuously developed for the urban system network in the HRB from 2006 to 2014.The scale of the urban system network in 2010 had been significantly expanded, which may have something to do with the continuous economic development of each urban node and the addition of many new nodes. 

The order degree of the urban system network declined slightly in 2010, indicating that the urban system network of the HRB had a better bearing capacity for these new nodes, and the urban system network did not appear more obviously complicated. The original urban system network was further developed, and the links between cities were strengthened from 2010 to 2014, which developed the efficiency of information transmission, the accuracy of information transmission, and the structure of the urban system network.

Generally speaking, the spatial development trend of highly connected areas in the HRB is from the central region to the eastern region and then to the western region. By 2014, the whole eastern, central, and northwestern regions had developed into highly connected areas, and the cities in the southwestern region were still not establishing links with other cities. The interaction between cities in this area and other cities is insufficient. Specifically, the area from here continues to expand to the northwest, and most of the original poorly connected areas had been transformed into highly connected areas. The areas of low EE and low QE are mainly concentrated in the northern and western regions; these nodes are in a more important position in the network structure, while most of the other nodes are in a more marginal position.

Based on the calculation of EE and QE of Urban System Network in Section 3.2.1 and Section 3.2.2, SE values of each city were obtained by using the calculation method of SE in Section 2.4. According to Natural Breaks (Jenks) classification, SE values were divided into five categories according to size, as shown in Figure 6.

As can be seen from Figure 6, from the perspective of EE and QE integration, Bozhou (low QE and high EE), Huainan (low QE and high EE) and Xuzhou (high QE and low EE), have the high SE in the whole network, with high efficiency of information exchange. Other types of areas have a trend of layer-by-layer distribution around the core of Xuzhou. However, the areas with low SE are mainly distributed far away from Xuzhou, to the southwest in HRB.

## 4. Discussion

### 4.1. Entropy of Urban System Network

Considering the spatial interaction relationship among cities, this study uses a gravity model method to establish an urban system network. On this basis, this study uses the entropy theory to analyze the urban system network structure. The urban system network developed and expanded with the development of time. The whole network has been in the process of continuous development and change. EE, QE, and SE can be used for state expression and analysis about the transfer of information of the entire network efficiency, urban system network structure, and the overall situation of the urban system.

In particular, the choice of threshold value for urban system network is very important which determines the sample selection of the urban system network. In this paper, in order to facilitate the visual expression and analysis of the urban system network, we set 4 as the threshold value for urban system network in HRB, that is, the connection (the interaction force) value should be higher than 4. Connection values between cities of less than 4 are temporarily ignored.

Entropy theory has been applied to the study of the urban system network and regional urban research. On this basis, this paper makes a new attempt to introduce entropy theory into the urban system network.

Entropy theory has been applied to the study of the urban system and regional urban research. On this basis, this paper makes a new attempt to introduce entropy theory into the urban system network. Efficiency Entropy(EE) refers to the uncertainty in the process of interaction between nodes. The EE of the spatial network of an urban system indicates the validity of the interaction of matter, energy, and information between cities. Quality Entropy (QE) describes the size of node interaction uncertainty. The QE of the spatial network of an urban system is another measurement of the effectiveness of the interaction of matter, energy, and information between cities. System Entropy (SE) is a comprehensive index, which integrates the value of QE and EE. The SE of a network structure is derived from the EE and QE of the network structure. So, EE, QE and SE of urban system network are calculated and analyzed in the urban system network in HRB.

As a whole, the HRB urban system network has been in the process of continuous development, and its internal nodes have been constantly changing. Generally speaking, the EE, QE, and SE of the urban system network are consistent with the development process of the urban system network, which also shows that our attempt is feasible. EE, QE, and SE can be used for state expression and analysis about the transfer of information of the entire network efficiency, urban system network structure, and the overall situation of the urban system.

The spatial structure of the urban system is a complex system. In order to study and analyze the spatial structure of the urban system, a model of urban system spatial network must first be established. There are many different ways to build an urban system network, such as using a radiation model, or neural network model. Considering the spatial interaction between cities, this study uses a gravity model to build an urban system space network. Economic aggregate data were used to describe the city development level. There are many methods available to describe the development of cities, such as night lighting data, foreign capital, number of enterprises, etc. In this study, linear distance was also used as the distance between cities, although the distance between cities can also be described by railway mileage, highway mileage, and other data.

### 4.2. Uncertainty Analysis and Improvement

As important urban economic data, the total economic data reflects the level of urban development to a certain extent. In this paper, the urban population and GDP statistics were used to determine the quality of the city. These data have some statistical errors. As an objective and effective method of data acquisition, remote sensing can reduce data error. There are many ways to describe urban development, such as foreign capital and number of enterprises. The selection of indicators and threshold analysis also need to be improved. For the study of urban systems, the selection of comprehensive indicators to determine the city level can better reflect the complex role between cities and regions. How to better describe the level of urban development remains to be further studied.

Most of the studies on regional urban systems focus on coastal cities with developed economies or rapid growth, or major inland cities, in locations such as Europe [12], Washington, D.C. [35], Beijing [36], Shanghai [37], Wuhan [38], and Shenyang [39]. Most of the research units are based on administrative divisions. The study of urban systems in China is focused on the Yangtze River Delta, Pearl River Delta, the Beijing–Tianjin–Hebei region, central-south Liaoning and other southeastern coastal areas, and the major supporting reform experiments in China. Systematic attention has not been paid to the complete geographical units in China. The HRB urban system is an important geographic transitional urban system in China, and there has been little research on its overall development.

The driving factors of development of the HRB urban system may be mainly related to natural factors (e.g., water system structure) [40]. Therefore, it is necessary to use other methods to conduct in-depth quantitative calculation and qualitative analysis of the spatial network structure of urban systems in river basins; such methods include Fractal [41],Cellular Automata Model [42], Support Vector Machine [43], Systematic Dynamics [44], Self-Organizing Structure Model [12], Generalized Equilibrium Model [13], Logical Model [45], etc.

The development of an urban system is complex system engineering, involving different spatial scales [46,47,48]. In addition to location, overall regional structure, and other related factors, the optimization of individual urban internal structure is closely related to the overall strength of regional urban system self-organization [49]. Further research will also focus on the expression, measurement, analysis, and scale conversion of urban spatial structure at different spatial scales.

## 5. Conclusions

In this study, the concepts of the gravity model and information entropy were introduced for the study of the spatial network structure of an urban system. Firstly, we used the gravity model to construct and express the spatial network structure of the urban system. Secondly, we used three kinds of network entropy models (EE, QE and SE) to calculate and show the urban system network structure. Finally, the spatial and temporal evolution of the urban system network structure in the HRB was analyzed. The conclusions are as follows:(1)The entropy theory based on the gravity model can express and analyze the network structure and its spatio–temporal patterns of the regional urban system. The QE, EE and SE of the spatial network of an urban system network indicate the validity of the interaction of matter, energy, and information between cities.(2)The spatial network structure of the regional urban system can be build by the gravity model, and the threshold value of the interaction force between cities of 4 is suitable for the construction of the spatial network of the urban system in the HRB.(3)The spatial and temporal distribution of the network structure of the urban system in the HRB is uneven. Generally speaking, the northern and eastern regions are superior to the western and southern regions. Spatially, the development of the urban system network in the HRB is unbalanced, showing a layer-by-layer spatial distribution centered on the core city of Xuzhou.

Additionally, it is necessary to put forward specific partition results for the interaction force value between cities, so that more cities can be analyzed on different spatial scales. Understanding the mechanisms driving the change of urban system network entropy needs to be combined with the actual situations in the region; this will be addressed in the next research phase.

## Figures and Tables

**Figure 1 entropy-21-00020-f001:**
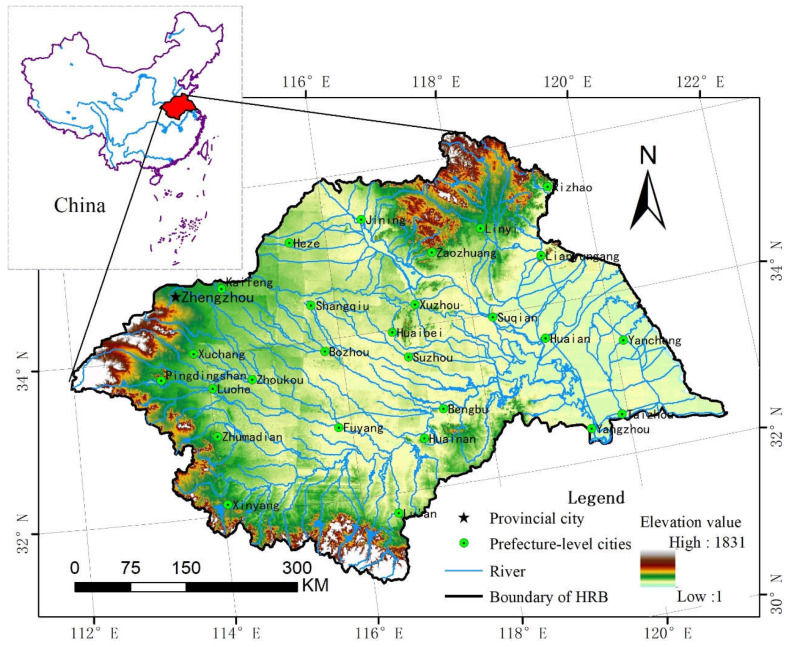
Study area.

**Figure 2 entropy-21-00020-f002:**
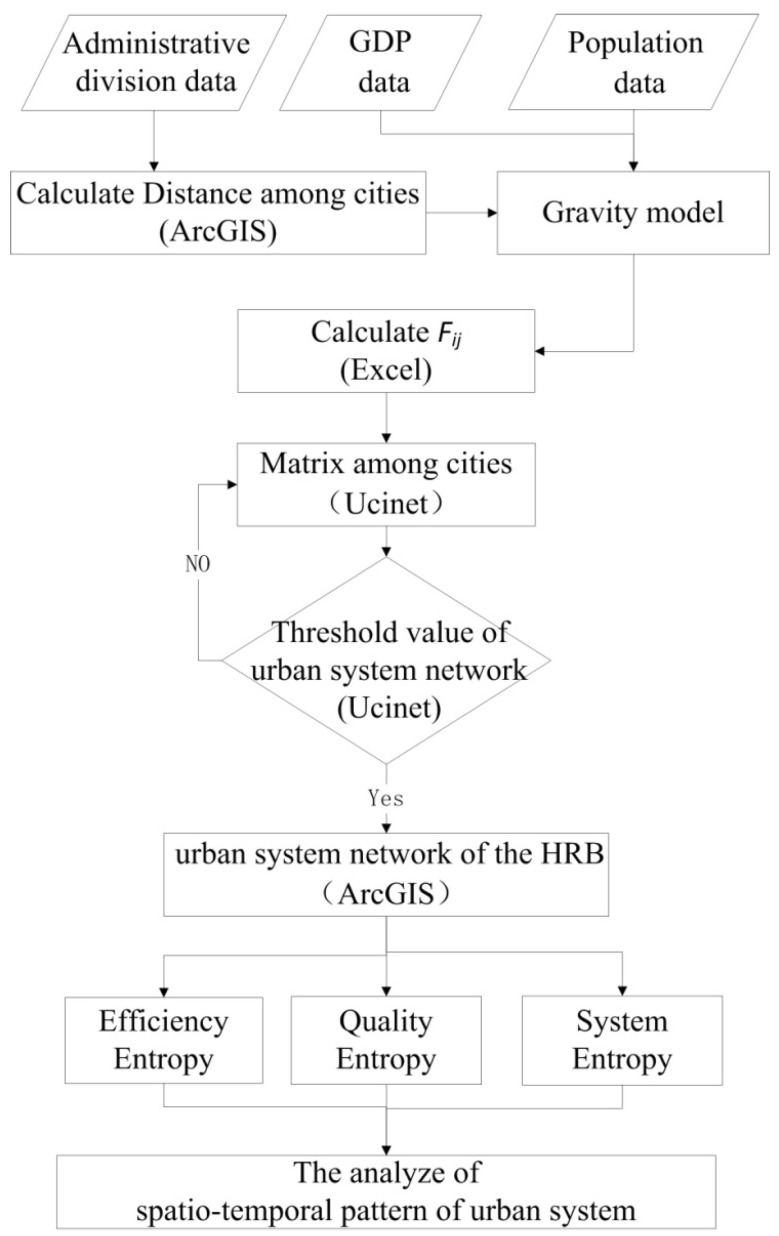
The flow of data processing and analysis.

**Figure 3 entropy-21-00020-f003:**
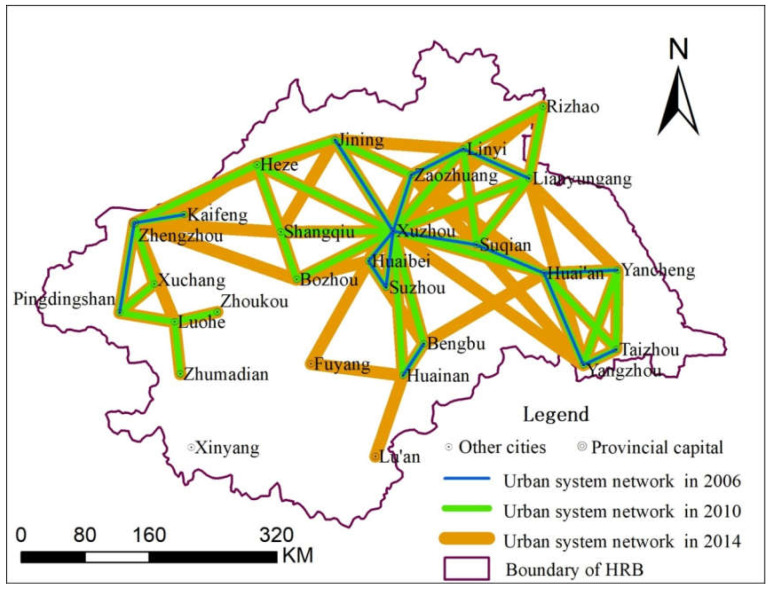
The urban system network of the HRB in 2006, 2010 and 2014.

**Figure 4 entropy-21-00020-f004:**
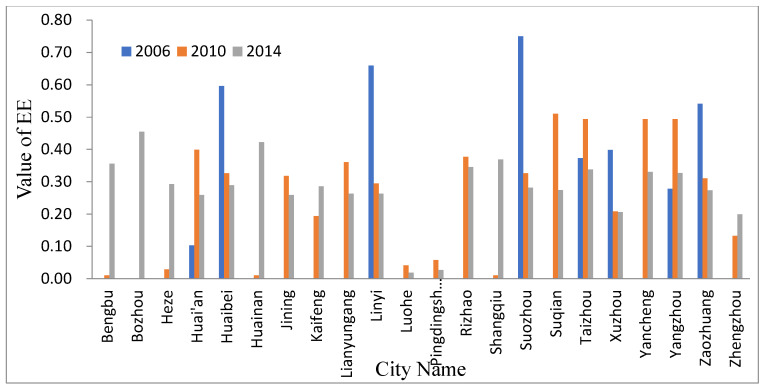
Efficiency Entropy (EE) of the urban system network of HRB.

**Figure 5 entropy-21-00020-f005:**
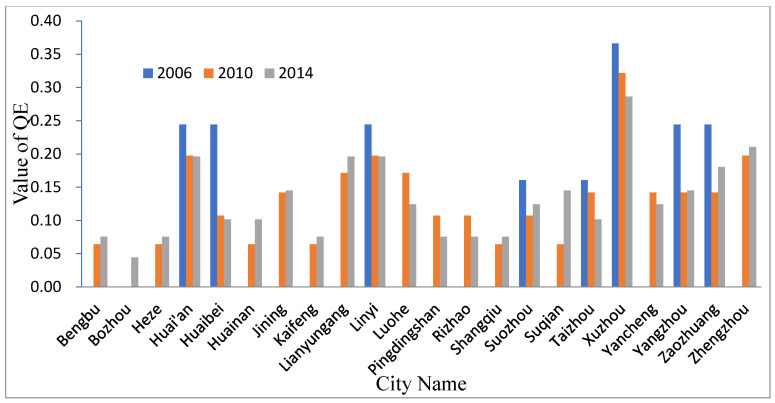
Quality Entropy (QE) of the urban system network of the HRB.

**Figure 6 entropy-21-00020-f006:**
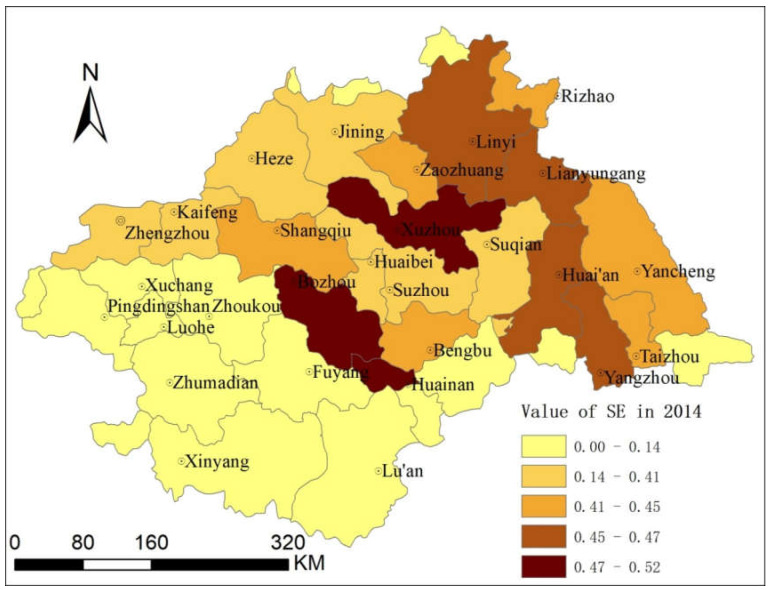
Entropy of the urban system in the HRB (2014).

**Table 1 entropy-21-00020-t001:** Efficiency Entropy (EE) of the urban system network in 2006, 2010, and 2014.

Year	Total Number of Connections	Total Micro-State	Maximum EE	EE	Order Degree
2006	45	86	6.43	5.34	0.17
2010	252	623	9.28	7.79	0.16
2014	528	1549	10.60	8.85	0.17

**Table 2 entropy-21-00020-t002:** Quality Entropy (QE) of the urban system network in 2006, 2010, and 2014.

Year	Total Micro-State	Maximum Structure Entropy	Structural Entropy	Order Degree
2006	18	4.17	2.75	0.34
2010	65	6.02	4.01	0.33
2014	105	6.71	4.15	0.38

**Table 3 entropy-21-00020-t003:** System Entropy (SE) of the urban system network in 2006, 2010, and 2014.

Year	H1 (EE)	H1m (EE)	R1 (EE)	H2 (QE)	H2m (QE)	R2 (QE)	H (SE)	R (SE)
2006	5.34	6.43	0.17	2.75	4.17	0.34	8.09	0.51
2010	7.79	9.93	0.16	4.01	6.02	0.33	11.80	0.50
2014	8.85	10.60	0.17	4.15	6.71	0.38	13.00	0.55

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
