# Peer review of "Spatio–Temporal Pattern of the Urban System Network in the Huaihe River Basin Based on Entropy Theory"

_entropy, 2018, doi:10.3390/e21010020_

Round 1

Reviewer 1 Report

The authors explore an interesting and timely topic that perfectly matches the scope of the journal. I think the paper would make an important contribution with proposing a method using different types of entropy to analyze spatio-temporal patterns between cities.

This study has good amount of discussion/conclusion about the results. However, I believe that the authors did not explain all of the aspects of the methodology thoroughly. Some examples:

-          It’s not clear why the authors used Eq. 3 in the study.

-          What are the meanings of H1, H1m and R1 in this study? What can we understand from largest EE and the order of network structure? (Section 2.2)

-           Same things for H2, H2m and R2 (Section 2.3)

-          Why the authors select 1 for α and β (Section 2.4)

-          The authors should discuss more about UCINET in p. 4

-          How did the authors define the connection value threshold? What is the range of this value? In addition, why did they pick 4 as their threshold?

Minor issues:

I advise the authors to get editing help from someone with full professional proficiency in English and solve some typos and punctuation issues.

Consistency problem in the figures:

Figures 1 & 3: There should be some similarity in making geographical maps in one paper – like the format of scale symbol, and north arrow

Figure 3: Why the north arrow is tilted?

Figures 4 & 5: Add Y label and also use the same color for each year.

Finally, I suggest that the authors add more details in methodology section and make this section stronger in terms of how they relate these elements to urban systems.

Author Response

Dear Reviewer:

We appreciate the valuable comments and suggestions of the reviewer, which have been fully considered in our amendment of the manuscript. During the past few weeks, we have considered very carefully about the questions and suggestions enclosed in the comments, and made major modifications to make this research more convincing, explicit and acceptable. Followings contain the detailed modifications and complementary explanations made in the revision according to your comments, questions and suggestions (red word). Point by point responses to the comments of reviewer are listed below (black word).

Reviewer 2 Report

In this research, spatio-temporal pattern of urban system networks in the Huaihe River basin is investigated using Entropy based approach. Basically, three types of Entropy (Efficiency, Quality and System) were calculated to evaluate urban system network consists of multiple cities. Also a Gravity model was used to investigate spatial pattern of GDP.

Please see my comments, questions and suggestions below:

1) Authors use a physically complete geographic unit (Huaihe River Basin) as their case study area to analyze mainly socio-economic relationships (GDP generations) among cities. An economic region does not have to perfectly fit to a physical unit. I strongly suggest authors to introduce how this study site is isolated from the surroundings.

2) In the introduced Gravity model, authors multiply population and GDP for a given city, then compute square root of the multiplication. In fact, GDP is a function of population. In economics, we generally compute GDP per capita. Because we know large population generates a large GDP. Therefore, we can rewrite GDP as GDP per capita (or in short GDP*). When we place this information into the introduced gravity model, square root of (GDP* x population x population) Then, what you get is actually equivalent to GDP per capita. I suggest simplifying their formulation of the gravity model.

3) Authors description of Efficiency and Quality Entropy on page 3 does not clearly explain how they computed values in the result section. For instance, Efficiency Entropy (EE) is computed based on the shortest distance between two nodes. If so, how EE values vary over time. The only possible explanation is to have different A1 each time point. Therefore, this section requires clarifications.

4) The purpose of introduced gravity model is not clear in the study. My understanding from the paper, the approach is used only for investigation of spatial pattern of GDP in the region. Also, temporal dimensions are examined for discrete time points. When I look at the results, they are bizarre. For instance, knowing the fact that Zhengzhou is a province, Xuzhou looks as a more important economic concentration area. Geographically, it's supposed to be, however providing GDP information for all cities and the province is a better idea to fix this uncertainty.

Author Response

Dear Reviewer:

We appreciate the valuable comments and suggestions of the reviewer, which have been fully considered in our amendment of the manuscript. During the past few weeks, we have considered very carefully about the questions and suggestions enclosed in the comments, and made major modifications to make this research more convincing, explicit and acceptable. Followings contain the detailed modifications and complementary explanations made in the revision according to your comments, questions and suggestions (red word). Point by point responses to the comments of reviewer are listed below (black word):

Reviewer 3 Report

See review report (attached pdf)

Author Response

(The authors gave the same response as above.)

Round 2

Reviewer 2 Report

After reviewing the resubmitted version with author's response to reviewers' comments, I don't see a significant improvement in the manuscript and I am not satisfied with the author's response in order to recommend this manuscript for publications. As described in my first review, the applied methodology should have been clearly explained in the manuscript.

Reviewer 3 Report

The modifications made by the authors to answer referee's remarks significantly improved the quality and scientific soundness of the paper, which seems now suitable for publication. Some minor modifications are suggested below; I do not need to see the paper again since these are mostly additional suggestions.

-  Literature review and purpose of the paper are more organized and relevant now. Some logical structure or organized typology may still lack but this is not a crucial issue.

- The addition of the data processing pipeline makes the procedure much clearer, and reproducibility is higher now.

- The answer on the remark for the gravity model is an interesting discussion, it could be integrated in the paper.

- Precisions on GDP and the Chinese context are fine.

- The section on entropies is also clearer now (also thanks to the missing reference that was added, even if I could not find it, it still backs up definitions).

- The justification of the area of study is still rather lightweight, we could expect a geographical contextualization (but this remains a minor issue not necessary relevant for the purpose of this paper, but that would be more crucial for a geography journal for example).

- The choice of the threshold value is not justified by a sensitivity analysis. It can be accepted as an arbitrary choice in this work but authors should keep in mind the role of sensitivity analyses and robustness assessment in future related works.

- Figures are much more readable now, and interpretations of results better justified and explained.

- Similarly, the reorganisation of the conclusion makes it much clearer.

- Concerning reproducibility, there are naturally different degrees of reproducibility; in the current state, thanks to the additional information on the data pipeline and entropy definition, it should indeed be reproducible by collecting the data and recoding the analysis (I do not have time to test that but I trust the authors). A higher level of reproducibility implies providing all data and code or files used to produce the figures, on an open repository using git for example (what can furthermore provide transparency on the research process).
